# Prompt Graft Cooling Enhances Cardioprotection during Heart Transplantation Procedures through the Regulation of Mitophagy

**DOI:** 10.3390/cells10112912

**Published:** 2021-10-27

**Authors:** Zhichao Wu, Jialiang Liang, Wei Huang, Lin Jiang, Christian Paul, Bonnie Lin, Junmeng Zheng, Yigang Wang

**Affiliations:** 1Department of Cardiovascular Surgery, Sun Yat-Sen Memorial Hospital, Sun Yat-Sen University, Guangzhou 510120, China; wuzc@ucmail.uc.edu; 2Department of Pathology and Laboratory Medicine, College of Medicine, University of Cincinnati, Cincinnati, OH 45267, USA; liangjl@UCMAIL.UC.EDU (J.L.); huangwe@ucmail.uc.edu (W.H.); yayajianglin@gmail.com (L.J.); paulca@ucmail.uc.edu (C.P.); Bonnielin8888@yahoo.com (B.L.)

**Keywords:** heart transplant, cardioplegic solution, ischemia-reperfusion, cardiac arrest, mitophagy

## Abstract

A complete and prompt cardiac arrest using a cold cardioplegic solution is routinely used in heart transplantation to protect the graft function. However, warm ischemic time is still inevitable during the procedure to isolate donor hearts in the clinical setting. Our knowledge of the mechanism changes prevented by cold storage, and how warm ischemia damages donor hearts, is extremely poor. The potential consequences of this inevitable warm ischemic time to grafts, and the underlying potential protective mechanism of prompt graft cooling, have been studied in order to explore an advanced graft protection strategy. To this end, a surgical procedure, including 10–15 min warm ischemic time during procurement, was performed in mouse models to mimic the clinical situation (Group I), and compared to a group of mice that had the procurement performed with prompt cooling procedures (Group II). The myocardial morphologic changes (including ultrastructure) were then assessed by electron and optical microscopy after 6 h of cold preservation. Furthermore, syngeneic heart transplantation was performed after 6 h of cold preservation to measure the graft heart function. An electron microscopy showed extensive damage, including hypercontracted myofibers with contraction bands, and damaged mitochondria that released mitochondrial contents in Group I mice, while similar patterns of damage were not observed in the mice from Group II. The results from both the electron microscopy and immunoblotting verified that cardiac mitophagy (protective mitochondrial autophagy) was present in the mice from Group II, but was absent in the mice from Group I. Moreover, the mice from Group II demonstrated faster rebeating times and higher beating scores, as compared to the mice from Group I. The pressure catheter system results indicated that the graft heart function was significantly more improved in the mice from Group II than in those from Group I, as demonstrated by the left ventricle systolic pressure (31.96 ± 6.54 vs. 26.12 ± 8.87 mmHg), the +dp/dt (815.6 ± 215.4 vs. 693.9 ± 153.8 mmHg/s), and the -dp/dt: (492.4 ± 92.98 vs. 418.5 ± 118.9 mmHg/s). In conclusion, the warm ischemic time during the procedure impaired the graft function and destroyed the activation of mitophagy. Thus, appropriate mitophagy activation has emerged as a promising therapeutic target that may be essential for graft protection and functional improvement during heart transplantation.

## 1. Introduction

During heart transplantation, the use of a cold cardioplegic solution for flushing and the storage of the donor heart is thought to protect the graft function by slowing metabolism. However, unlike mouse hearts (whose temperatures can be immediately dropped to 2 °C), large animal hearts usually take 10–15 min to reach 4 °C during the procedure, despite protocols that include a flush with cold preservation solution via the aorta root cannula, and immediate ice cooling [1]. Thus, warm ischemic (WI) time is inevitable in the clinical setting.

It is generally accepted that the metabolic rate is the difference between WI and cold ischemia (CI) and, increasingly, the evidence indicates that CI and WI can result in different histological characteristics in lung, liver, and kidney preservation [2,3,4]. Iskender et al. [5] indicate that, even though long-term CI and short-term WI had similar severities of injury and cell death during lung preservation, the tissue inflammation levels were decreased, and the ATP levels were increased, during CI. Thus, a reduction of the warm ischemia time could minimize inflammation and reperfusion injury in the heart graft. However, the underlying mechanism of how CI protects grafts, and how WI impairs graft function during heart transplantation remains unclear, and further investigation is needed to optimize the protocols for clinical transplantation use. 

As the intracellular organ that continuously provides ATP to cells with high-energy requirements, mitochondria control the fate of cardiomyocytes by regulating the reactive oxygen species (ROS) levels, calcium storage, metabolic substrate utilization, etc. [6,7]. Thus, mitochondrial quality is critical for cell survival. ROS are also significantly increased in damaged mitochondria, which can further induce damage in mitochondrial proteins and DNA, forming a negative feedback loop. Mitophagy is a special type of autophagy in which the damaged mitochondria are removed, providing a critical mechanism for maintaining mitochondrial quality [6,8]. Cardiac mitophagy is necessary for cardiomyocyte recovery from the mitochondrial damage caused by ischemia or other factors [9,10,11,12]. Nevertheless, whether mitophagy is activated during CI, and whether the WI time during procurement has any effect on mitophagy, are still unknown.

In the present study, we examined whether the inevitable WI time during procurement has any significant impact on graft function. Moreover, we explored the potential mechanism of the injuries caused by the WI time on graft cold storage. We found that WI time is a key player in causing damage to the graft structure and function during heart transplantation. Mitophagy is widely activated during cold storage, but the WI time destroys mitophagy activation. Identification of the underlying mechanism of injuries caused by the WI time during procurement may contribute to the understanding of graft protection, which could further provide more relevant methods for prolonging the isolated heart preservation time.

## 2. Materials and Methods

### 2.1. Animals

Male and female C57BL/6 mice aged 8–12 weeks, and weighing from 20–30 g, were used in this study. The original generation of these mice was purchased from Jackson Laboratory (Indianapolis, IN, USA), and they were maintained and bred in the Division of Laboratory Animal Resources at the University of Cincinnati Medical Center. All research protocols conformed to the Guidelines for the Care and Use of Laboratory Animals, published by the National Institutes of Health (National Academies Press, eighth edition, 2011). All animal-use protocols and methods of euthanasia (pentobarbital overdose followed by thoracotomy) used in this study were approved by the University of Cincinnati Animal Care and Use Committee.

### 2.2. Materials and Instruments 

Histidine-tryptophan-ketoglutarate (HTK) solution, purchased from CUSTODIOL, was used as a cardiac preservation solution for this study. Heparin sodium injection (NDC-25021-400-30, Sagent Pharmaceuticals, USA) was used for the systemic heparinization of donors. An isoflurane vaporizer (Classic T3, Surgivet, Minneapolis, MN, USA) was used for the recipient anesthesia. A microsurgical microscope (OPMI, f170, Carl Zeiss, Jena, Germany) was used with 4, 10, 16, and 25× interchangeable magnifications. The 11-0 surgical sutures (Lingqiao, Ningbo, China) were used for vascular anastomosis, and 6-0 sutures (SUT-14-1, Roboz Surgical, Gaithersburg, MD, USA) were used for donor heart ligations and recipient vessel blockings during the surgery. We used 6-0 sutures (K801, Ethicon, Bridgewater Township, NJ USA) for the purse-string suture, and 4-0 sutures (C054D, Ethicon, Bridgewater Township, NJ, USA) were used for closing abdomens. Graft function was measured by Mikro-Tip pressure-catheter transducers (SPR-839, Millar Instruments, Houston, TX, USA) and analyzed with LabChart Reader (V8.1.14, ADInstruments, Sydney, Australia). 

### 2.3. Anesthesia

A mixed solution with 20% xylazine hydrochloride (20 mg/mL, NDC59399-110-20, AnaSed, IL, USA), and 80% ketamine hydrochloride (100 mg/mL, NDC11695-0702-1, Henry Schein, OH, USA), was diluted 10 times using 0.9% saline solution. Intraperitoneal injection was performed on both donor and recipient mice at a dose of 0.1 mL per 10 g body weight. Inhalation anesthesia (1% isoflurane, Covetrus, Melville, NY, NDC:11695-6777-2) and 6–8 L/min oxygen were delivered to the recipient via a mask from the anesthesia device system for increased oxygen supply and to prevent awakening during the surgery.

### 2.4. Surgery

0.3 mL of 100 Unit/mL heparin sodium was slowly injected into the donor mice at the inferior vena cava (IVC) using an insulin syringe. After waiting for 15 s for systemic heparinization, the chest was opened. Sterilized ice was then put into the chest, and the heart was wrapped with a 4 °C wet surgical gauze. The donor mice were randomly divided into two groups: (a) Group I (*n* = 22): in which we freed and ligated the superior vena cava (SVC), inferior vena cava (IVC), and azygos veins, cut off the IVC, SVC, azygos vein, aorta, and pulmonary artery, then ligated the pulmonary veins and removed the heart into 4 °C HTK solution, perfused the heart with cold HTK solution until there was no blood left, and kept the heart in 4 °C HTK solution for 6 h until transplant; and (b) Group II (*n* = 24): in which we cut off the aorta and pulmonary artery immediately, using 4 °C Bretschneider-HTK cardioplegic solution. The grafts were perfused until the heart completely stopped beating. Then, the superior vena cava (SVC), inferior vena cava (IVC), the azygos vein, and the pulmonary veins were ligated, followed by the removal of the heart into 4 °C HTK solution, and perfusion with cold HTK solution again until there was no blood left. The hearts were kept in 4 °C HTK solution for 6 h until transplantation. (Appendix A) 

The recipient operation was performed as previously described [13], with details of the procedure shown in Appendix A. Briefly, the abdomen of the recipient was opened with a midline incision, and a length of the abdominal aorta and IVC between the renal vessels and iliac bifurcation was freed. We placed 6-0 silk under both sides of the freed abdominal aorta and the IVC. The spinal veins were then ligated with 6-0 silk, and a slipknot was created to block the flow of blood to the abdominal aorta and the IVC. Finally, the donor’s ascending aorta and pulmonary artery were anastomosed to the recipients’ abdominal aorta and inferior vena cava, respectively, using 11-0 sutures.

### 2.5. Warm Ischemia Time Record during Procurement 

The warm ischemia time during the procurement process was recorded in the two groups and was defined as the period of time from opening the thoracic cavity to when the hearts were kept in a hypothermic condition without beating. 

### 2.6. Ultrastructure Measurement

Transmission electron microscopy was used to measure the ultrastructural changes of the donor hearts, which were procured using two different methods. After six hours of HTK preservation, the samples were fixed in 2.5% glutaraldehyde, then prepared for transmission electron microscopy, according to standard protocols [14]. Ultrastructural changes of the heart were analyzed at 3000× or 8000× magnification. Mitochondrial damage was then assessed using Flameng mitochondrial scores [15], as follows: Score 1: Broken cristae with a ruptured mitochondrial membrane; Score 2: Broken cristae with matrix clearing but membrane intact; Score 3: Swelled mitochondria with cleared matrix but intact cristae and membrane; Score 4: intact mitochondria; Score 5: intact mitochondria with mitochondrial granules. The mitochondrial area was used to measure swelling, and was quantified using Image J (NIH, Bethesda, MD, USA). At least 100 mitochondria from 5 randomly selected images were analyzed.

### 2.7. Histological Staining

Samples from the two groups were collected after 6 h of HTK preservation and fixed in 10% neutral buffered formalin solution (Sigma, St. Louis, MO, USA). Tissue blocks were embedded in paraffin for sectioning, and the sections were stained with hematoxylin-eosin (H & E). 

### 2.8. Western Blot and Band Quantitation

Protein samples were collected from the two groups after 6 h of HTK preservation. A total of 25 micrograms of protein per sample was loaded on a precast protein gel (Nupage, WG1401BX10, USA), followed by semi-dry Western blotting before transfer onto polyvinylidene fluoride (PVDF) membranes. The membranes were then blocked with 5% milk in TBST for 1 h and incubated overnight with primary antibodies against LC3 I/II (1:1000, rabbit, CST), SQSTM1/P62 (1:1000, mouse, Abcam), and GAPDH (1:1000, rabbit, CST). Subsequently, the membranes were incubated with HRP-conjugated secondary antibodies (1:10000, anti-rabbit or anti-mouse, CST) for 1 h at room temperature. Finally, the bands were visualized on immunoblots using ECL Chemiluminescent Substrate (Thermo Fisher, Waltham, MA, USA), and quantification of the band intensity was determined using Image J software (NIH, Bethesda, MD, USA).

### 2.9. LDH and CK Activity Assay

Grafts were preserved in 2 mL 4 °C HTK solution for 6 h, after which the HTK solution from the two groups was collected. Lactate dehydrogenase (LDH) activity in the HTK solution was measured using an LDH Activity Assay Kit (Sigma-Aldrich, MAK066, St. Louis, MO, USA). Creatine kinase (CK) activity in the HTK solution was measured using a CK Activity Assay Kit (Sigma-Aldrich, MAK116, MO, USA). All procedures followed the manufacturers’ technical instructions as indicated. 

### 2.10. Rebeating Time and Beating Score

The beating score of the heart grafts was assessed visually 10 min after restoring blood flow (before closing the abdomen), using the Stanford cardiac surgery laboratory graft scoring system [16] (0: No contraction, 1: Contraction barely visible or palpable, 2: Obvious decrease in contraction strength, but still contracting in a coordinated manner, 3: Strong coordinated beating but noticeable decrease in strength or rate, 4: Strong contraction of both ventricles, regular rate). The rebeating time was recorded as the time from when the blood block was released until ventricular contraction restoration. 

### 2.11. Left Ventricle Function Measurements 

Left ventricle systolic pressure (LVSP), left ventricle end-diastolic pressure (LVEDP), the maximal rate of change of the left ventricular pressure (±dp/dt), and the heart rate were recorded after 10 min of restoring the sinus rhythm, using a Millar pressure–volume (PV) catheter, and were analyzed using LabChart reader, as previously described [17]. Briefly, 6-0 sutures were used to make a pursing string suture on the donor heart’s apex, then a 22 G needle was used to make a hole in the middle of the purse. The tip of the Mikro-Tip pressure catheter was placed into the left ventricle via the hole, and the purse-string suture was tightened. LabChart software was used to record and analyze the data once heart-beating was stable.

### 2.12. Rapamycin Treatment and Survival Analysis 

Rapamycin (A10782, AdooQ Bioscience, Irvine, CA USA) was dissolved according to the manufacturer’s instructions (2% DMSO, 30% PEG300, 5% Tween80, ddH2O. An amount of 8 mg/kg [18] rapamycin, or the same volume vehicle, was administered via intraperitoneal injections at 12 h prior to graft procurement on the donor mice in Group I. Donor hearts were procured and stored in 4 °C HTK solution for 12 h before transplantation of the hearts to the recipients. The transplanted hearts were touched every day, and the survival curve for the donor hearts was recorded.

### 2.13. Statistical Analysis

All results are expressed as the mean ± SD. Comparisons between the two groups were performed by using T-test analysis in SPSS 17.0. A *p* value < 0.05 was considered significant. 

## 3. Results 

### 3.1. Warm Ischemic Time Is Reduced by Immediate Perfusion with Cold Cardioplegic Solution

Because of the size and instrument limitations, WI is inevitable in mouse heart transplantation models. Once the thoracic cavity has been opened, effective breathing stops immediately, and the donor’s heart suffers WI damage. However, it is reported that cold perfusion can be used to quickly drop the mouse heart temperature to 2 °C [1]. Thus, in Group I, we procured the heart with a delayed perfusion of cold cardioplegic solution, which caused a 10–15-min warm ischemia that accurately mimics the conditions found in the clinical setting. In Group II, we performed a prompt and complete cold cardioplegic solution perfusion before procurement in order to minimize the WI time. (Appendix A and Figure 1A).

The WI time of the cardioplegic solution treatment in Group II was around 10 min shorter than that of the control treatment in Group I (182.9 ± 42 vs. 776.9 ± 130 s) (Figure 1B), contributing to the morphological differences between the two groups. For the recipient operation, there were no significant differences in the blocking times between groups, which excluded the effect of the recipient surgery time on graft function (Figure 1C).

### 3.2. Graft Mitochondrial Integrity Is Maintained by Prompt Cardiac Cooling

After 6 h of preservation, the heart graft was collected for transmission electron microscopic examination. Isolated hearts procured without a prompt cardiac arrest (Group I) showed a disordered sarcomere and mitochondria arrangement, and the continuity of some sarcomeres was disturbed (Figure 2A).

Moreover, contraction bands indicating myocardial injury were observed in this group (as shown by the yellow arrows), and the majority of the mitochondria were swollen with matrix clearance. Thus, while the mice in Group I showed damaged mitochondria with incomplete mitochondrial membranes and broken cristae, the grafts of the mice in Group II showed sarcomere and mitochondria orders that were smooth and that displayed a well-organized ultrastructure with intact membranes and cristae (Figure 2A), and no contraction bands were observed. Interestingly, mitophagosomes were absent in the grafts from Group I, but were observed in the Group II grafts (as shown by the white arrows). Moreover, the average Flameng score of the mice in Group II was higher than that of Group I (Figure 2B), while the mitochondrial area was significantly smaller in Group II than in Group I (Figure 2C).

### 3.3. Prompt Cardiac Cooling Protects Graft from Ischemic Injury during Preservation 

Histological changes in the two groups were assessed after 6 h of preservation in HTK solution. H & E staining showed an abnormal linear arrangement, with widely inter- and intracellular edema in Group I (Figure 3A).

However, the grafts in Group II showed a linear arrangement and the edema was alleviated. LDH and CK leakage were then used as a measure of tissue damage after 6 h of cold preservation in HTK solution. The LDH and CK activity in the HTK solution were significantly less in Group II (Figure 3B,C), indicating less severe damage in this group.

### 3.4. Prompt Cardiac Cooling Enhances Graft-Beating Recovery after Transplantation

After 6 h of cold ischemia preservation in solution, syngeneic graft implant surgery was performed on the mice from each group (Appendix A). When the blood flow of the graft was restored, the grafts in Group I showed a significant ischemia-reperfusion injury, as evidenced by an irregular red and white tissue color (Figure 4A), while all of the grafts collected from Group II presented a distributed red color. 

The rebeating times and the Stanford beating scores were further applied to measure the cardiac status in mouse heart transplant models. As compared to Group I, Group II showed a shorter rebeating time and a higher beating score (Figure 4B,C), indicating less cardiac ischemia injury to the graft. Group II also showed a quicker response to blood-flow restoration. 

### 3.5. Prompt Cardiac Cooling Brings Stronger Cardiac Contractility

To accurately measure the ventricle function and quantify the damage to the graft function caused by 10 min of ischemic heart-beating, a pressure catheter was used to directly detect the left ventricle pressure of the graft after 10 min of restoring the sinus rhythm. The left ventricle systolic pressures (LVSPs), the left ventricle end-diastolic pressures (LVEDPs), the ±dp/dt, and the heart rates of the grafts were recorded. The grafts in Group II showed a higher-pressure peak and a pressure-changed-rate peak, as compared to Group I (Figure 5A). 

There was no significant difference in the heart rate between the two groups (Figure 5B). We used the first nine waves that appeared after the stable curves for the function analysis. The LVSPs in Group II (31.96 ± 6.54 mmHg) were significantly higher than that of the grafts in Group I (26.12 ± 8.87 mmHg) (Figure 5C). However, there was no significant difference in the LVEDPs between the two groups (Figure 5D). Furthermore, the +dp/dt max, which indicates the maximum pressure growth rate, was significantly higher in Group II than in Group I (Figure 5E). The faster pressure rise reflected a better contractile function in Group II. Similarly, the -dp/dt max, reflecting the maximum pressure decay rate, was significantly higher in Group II than in Group I (Figure 5F). Thus, ten minutes of ischemic beating time can cause a significant loss of cardiac contractility. 

### 3.6. Mitophagy Is Enhanced with Prompt Perfusion of Cold Cardioplegic Solution 

Since mitophagosome formation was commonly observed in Group II under transmission electron microscopic examination, further tests were performed to confirm the activation of mitophagy. A Western blot analysis of the mitophagy-related proteins, such as LC3-II and SQSTM1(P62) [19,20], indicated a decrease in the SQSTM1 (P62) levels, indicating the lysosomal degradation of the autophagosomes in Group II (Figure 6A,B).

Consistent with electron microscopy, the expression of the cleaved LC3-II was significantly increased after 6 h of cold solution preservation in Group II, as compared to Group I (Figure 6A,C,D). 

### 3.7. Activation of Mitophagy Prolongs Isolated Heart Preservation Time

To further verify that activation of mitophagy prolongs the preservation time for hearts after WI injury, rapamycin (which has been demonstrated to stimulate mitophagy [21]) was administrated to the mice in Group I before procurement. The graft implantation surgeries were performed after a prolonged cold storage time (12 h), and the recipients who accepted a vehicle-treatment donor heart showed serious symptoms of systemic inflammatory response syndrome (eye secretions, hypothermia, inactivity). These mice had weak donor heart-beating, and all of the donor hearts died within three days post-surgery. In contrast, the recipients with rapamycin-treated donor hearts showed an improvement in the survival rate (Appendix A). 

## 4. Discussion 

At present, hypothermic immersion preservation is still the gold standard for isolated heart preservation. However, the storage time for isolated hearts is limited to 4 to 6 h using this method [22,23]. Prolonged preservation times increase perioperative mortality [23] and introduce time limitations that further emphasize the shortage of donor organs. Several novel preservation technologies have been used in the past in an attempt to minimize this problem, including continuous perfusion preservation, intermittent perfusion, high-pressure gaseous mixture preservation, and supercooling preservation. However, outcomes are still not optimized in the clinical setting. Cold cardioplegic solution perfusion and ice cooling are generally used during the heart preservation process as an effective protection for avoiding warm ischemia. However, the size of a large animal heart may prevent the core temperature from lowering to 4 °C in a time-sensitive manner. Thus, the WI time during donor heart isolation on large animals is inevitable, and this slow-cooling period of time has been shown to lead to a significant succinate accumulation in the mitochondria [10]. 

Previous studies have been based on the concept that the mechanisms for warm ischemic (WI) injury are similar to that of cold ischemic (CI) injury during heart transplantation in rodents. However, developing research has demonstrated that the underlying mechanisms of WI and CI may differ. For instance, Hansen et al. [24] have shown that some agents that can protect the rat heart from WI reperfusion injury (such as arginine and indomethacin) fail to have similar results with CI injuries. In contrast, other agents (such as deferoxamine mesylate) confer cardiac protection on both WI and CI injury. This finding was the genesis of our hypothesis that the WI time during the graft procedure can initiate a different underlying pathological process than occurs during CI. In our study, transmission electron microscopy was used to verify this hypothesis through the examination of the ultrastructural differences between the two groups. As expected, a large number of mitophagosomes were observed in the grafts processed with prompt cardiac cooling. In addition, the expression difference of LC3/II and p62 between the two groups further confirmed the elevated mitophagy activation in Group II. 

The heart is highly dependent on energy provided by mitochondrial oxidative phosphorylation. Mitochondria play an important role in maintaining essential cellular processes, including metabolite synthesis, calcium storage, and controlling the intracellular survival and death pathways [25]. Cardiac mitochondria also play a role as the mediator of cell survival and death following heart transplantation. Therefore, preventing mitochondrial dysfunction induced by acute myocardial ischemia/reperfusion (I/R) is an important protective strategy for the heart graft, and optimizes mitochondrial function to provide energy for maintaining cell membrane ion pumps, keeping the cell membrane intact, and supplying the energy required for cardiac excitation-contraction coupling. Our results showed mitophagy was active in Group II, suggesting that the appropriate regulation of mitophagy is essential for isolated graft protection. The accumulating evidence has reported that mitophagy is essential for protecting the heart in the myocardial infarction (MI) model and that mitophagy dysregulation can induce cardiomyocyte death in MI models [26]. Our data represent the first time that appropriate mitophagy has been linked with a beneficial adaptive response to protect against I/R injury during heart transplantation. Our data further demonstrate that heart grafts tolerated ischemic injury by using a mitophagy stimulator (rapamycin), which is consistent with other reports indicating that mitophagy is a key player in maintaining mitochondrial quality and, thus, conferring cardioprotection against stress [27]. Mitophagy has, therefore, emerged as a promising therapeutic target that may be essential in determining when and how heart transplantation patients are protected from I/R injury.

Mitochondrial permeability transition (MPT) is represented by a serious swelling of the inner membrane in the mitochondrial matrix that distends the outer membrane until it ruptures [28,29,30]. Petit et al. [28,29] have demonstrated that the irreversible permeability transition induced outer mitochondrial membrane disruption, leading to the release of cytochrome C and apoptosis-inducing factor (AIF). This step is closely related to apoptosis induced by the mitochondrial pathway. We have demonstrated that the WI time during the graft procedure results in mitochondrial swelling, which further leads to cell apoptosis. Thus, we concluded that mitochondria experience a different pathological process due to the prolonged WI time during cold storage. Moreover, as a part of the inner membrane, cristae deeply extend into the matrix and contain the majority of ATP synthase, fully assembled complexes of the electron transport chain, and the small soluble electron carrier protein, cytochrome c [31,32]. Cristae are the main site where the mitochondrial function is performed, and the results indicate that a prompt cooling induction can protect mitochondrial cristae density. In contrast, the density of the cristae was significantly lower after several minutes of warm ischemia. Therefore, we have proposed that one of the main benefits of prompt cooling is the maintenance of the cristae density during CI. 

Furthermore, the WI time was the main difference between the two experimental surgical methods. The results from our study show that even a 10-min WI time is sufficient to make a difference in the pathological mechanism and cardiac function. This is consistent with other reports of reduced survival in heart transplant recipients with increased WI times [33]. In fact, the WI caused by the slow cooling process during graft procurement not only affects mitochondrial function, but also affects the underlying mechanisms of transplant immunity. A prolonged WI time can cause both adaptive and innate responses [34,35]. Thus, we suggest that optimizing the WI time during heart transplantation is critical for optimizing outcomes for post-heart-transplantation surgery. Furthermore, since the inevitable WI time destroys mitophagy (which protects mitochondria during cold storage), the activation of mitophagy may be a potential method to save graft function and prolong the isolated heart preservation time in the clinical setting.

The primary pitfall of this study is that the rodent heterotopic heart transplantation model lacks a preloaded volume. On the basis of the Frank–Starling mechanism, cardiac contractility is positively correlated to the effective preload. Because of the limitations of this model, the only preloaded volume of the left ventricle is from aortic regurgitation. Here, a new method for measuring cardiac function was applied in which a pressure catheter was used to measure the graft function more accurately than echocardiography by completely unloading the ventricle. Thus, the LVEDPs in each group approach 0 mmHg. Although the cardiac function is comparable between each group, the unloaded status may differ from a true clinical setting.

In conclusion, this is the first study demonstrating that the activation of mitophagy improved isolated heart preservation, and that the WI time, resulting from a slow cooling process, can destroy mitophagy activation and affect the graft functional recovery. The present results suggest that the inevitable WI time during graft procurement in the clinical setting may lead to a different pathophysiological process in the cardiac graft during cold storage, which may be an essential factor for limiting the isolated heart preservation time. Aiming for appropriate mitophagy activation may provide a new therapeutic to further protect graft function and prolong the isolated heart preservation time. Moreover, although it is widely accepted that prompt cardiac arrest is still an essential and protective process to cardiomyocytes during cardiopulmonary bypass surgeries, the underlying mechanism of this protection remains incomplete. Our study provides a theoretical basis for the importance of appropriating mitophagy activation during warm-ischemia time in tolerating I/R injury. 

## Figures and Tables

**Figure 1 cells-10-02912-f001:**
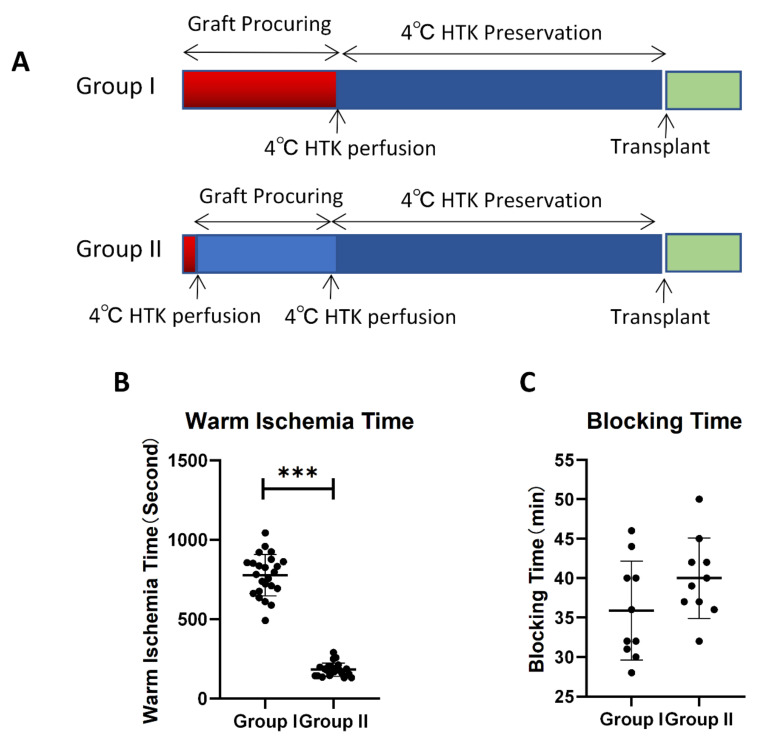
Overall evaluation of two surgical procedures. Schematic surgery process: (**A**) of two groups. Blocking time warm-ischemic time; and (**B**) blocking time (**C**) of two groups ((**B**): *n* = 23, (**C**): *n* = 10, *** *p* < 0.001).

**Figure 2 cells-10-02912-f002:**
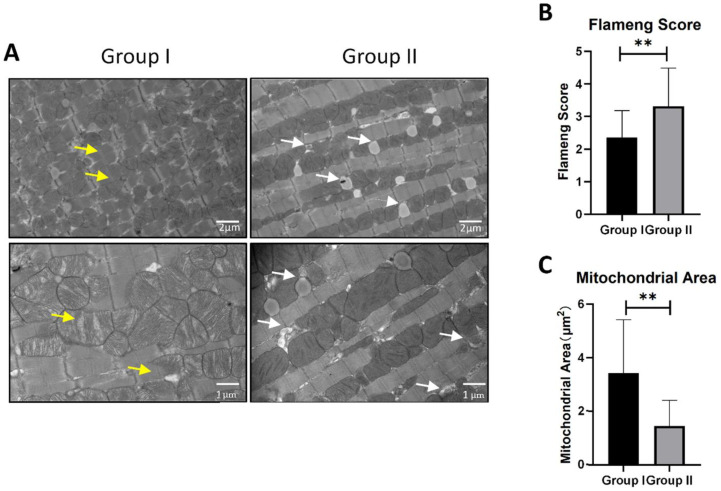
The effect of two surgical procedures on mitophagy induction after 6 h HTK preservation. (**A**) Ultrastructure changes under transmission electron microscope image of two groups. Yellow arrows show contraction bands and damaged mitochondria with matrix clearance and ruptured membrane. White arrows show mitophagy. Scale bars respectively indicate 2 μm and 1 μm. (**B**) Mitochondrial injury was measured by the Flameng score system (*n* = 3). (**C**) Mitochondrial area was analyzed by Image J. At least 100 mitochondria from 5 randomly selected images were analyzed (*n* = 3). ** *p* < 0.01.

**Figure 3 cells-10-02912-f003:**
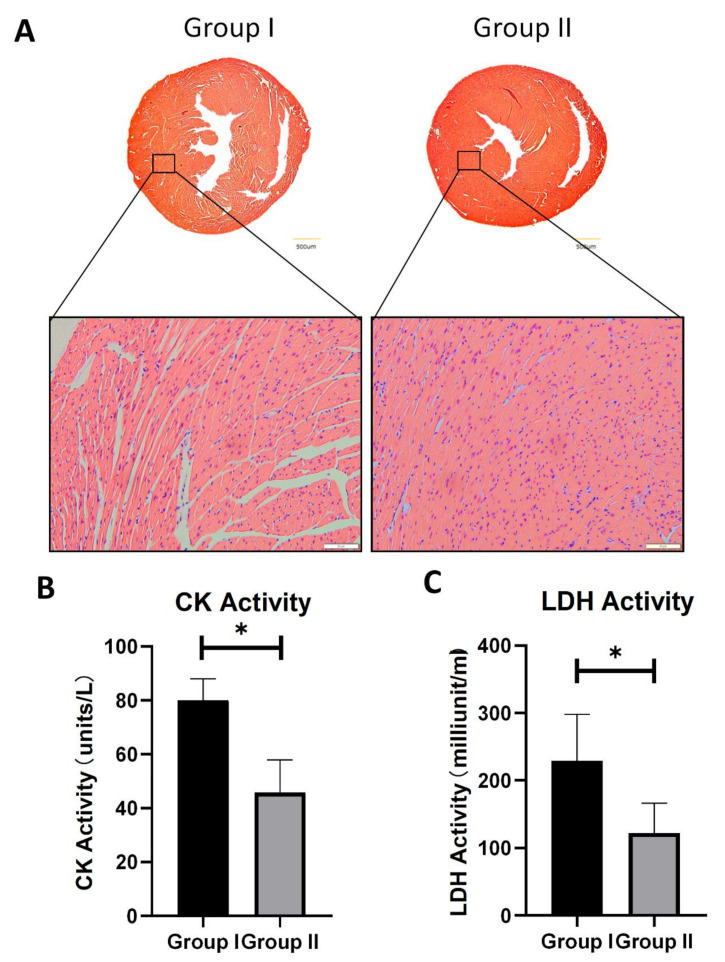
Heart damage evaluation of two groups after 6 h HTK preservation. (**A**) Representative H&E staining of two groups. Scale bar indicates 500 μm and 50 μm, respectively. (**B**) Creatine kinase activity level in HTK solution of two groups after 6 h preservation. (**C**) LDH activity level in HTK solution after 6 h preservation. N = 4–5, * *p* < 0.05.

**Figure 4 cells-10-02912-f004:**
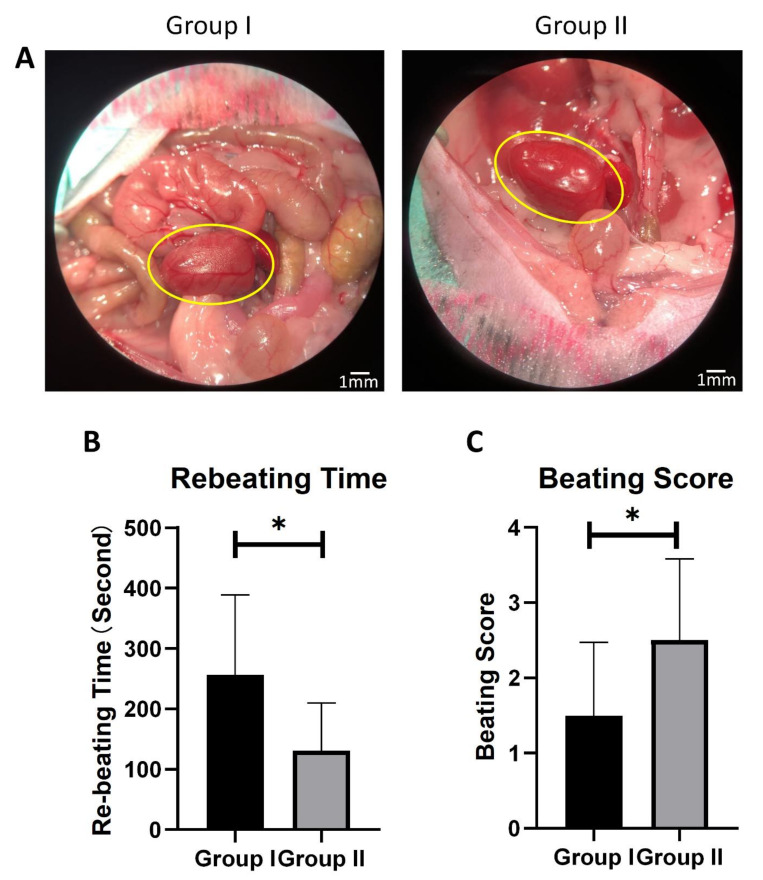
Graft functional recovery from 6 h cold ischemia in two groups. (**A**) Representative reperfusion photographic images of two groups after 6 h preservation. Scare bar indicates 1 mm. (**B**) Donor heart rebeating time during reperfusion. (**C**) Beating score measured by the Stanford cardiac surgery laboratory graft scoring system. *n* = 10, * *p* < 0.05.

**Figure 5 cells-10-02912-f005:**
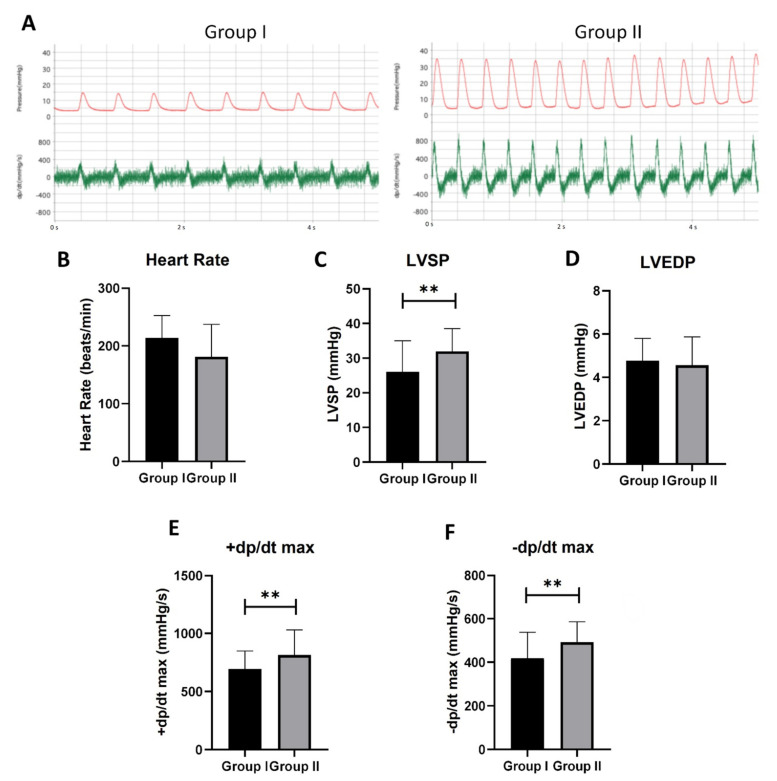
Graft function analyzed by Millar pressure catheter. (**A**) The representative LabChart images of Groups I and II. The red wave represents the left ventricle pressure (mmHg). The green wave represents the ±dp/dt (mmHg/s). (**B**) Heart rate (*p* = 0.309). (**C**) Left ventricle systolic pressure (LVSP). (**D**) Left ventricle end-diastolic pressure (LVSP) (*p* = 0.439). (**E**) The maximum dp/dt in the systolic period. (**F**) The maximum dp/dt in the diastolic period. The first nine waves of each sample were measured. *n* = 5, ** *p* < 0.01.

**Figure 6 cells-10-02912-f006:**
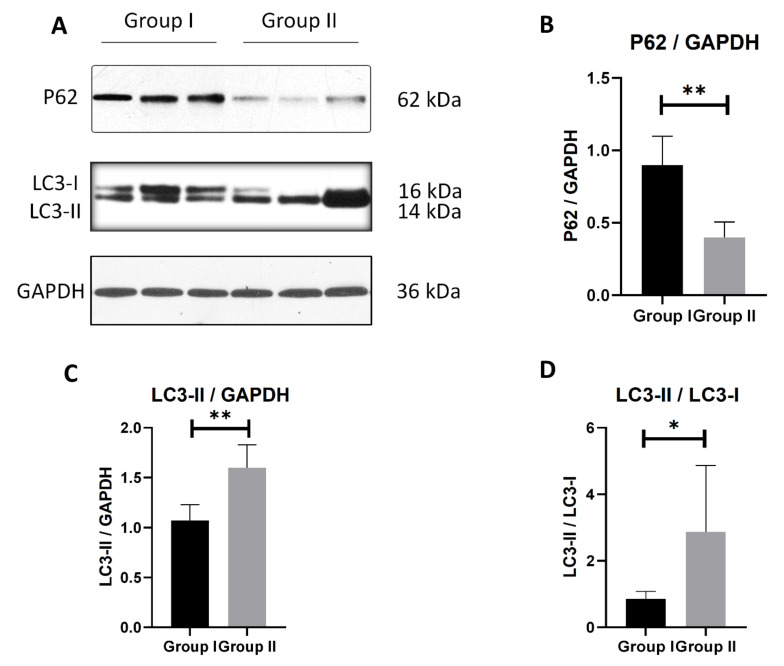
Representative immunoblots of mitophagy-related protein. (**A**) P62, LC3-I, LC3-II protein expression levels of the two groups. (**B**) Corresponding quantification of p62 to GAPDH. (**C**) Corresponding quantification of LC3-II to GAPDH. (**D**) Corresponding quantification of LC3-II to LC3-I. *n* = 6, * *p* < 0.05, ** *p* < 0.01.

## Data Availability

Not applicable.

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
