# Peer review of "Prompt Graft Cooling Enhances Cardioprotection during Heart Transplantation Procedures through the Regulation of Mitophagy"

_cells, 2021, doi:10.3390/cells10112912_

Round 1

Reviewer 1 Report

The present article entitled “Prompt Cardiac Arrest Enhances Cardioprotection During Heart Transplantation Procedures Through the Regulation of Mitophagy” presented by Zhichao Wu and colleagues is quite simple, well presented and offer a simple view of the use of a cardioplegic solution while inducing prompt cardiac arrest is more clinically relevant, and activation of mitophagy is essential for isolated graft protection.

Remark 1 references

In the following paragraph the authors have to indroduce the princep articles

The authors have writen “Mitochondrial permeability transition (MPT) is represented by a serious swelling of the inner membrane in the mitochondrial matrix that distends the outer membrane until it ruptures [28] (This reference could simply be replaced by the reference 3 below that is the original). As a result of MPT, the ruptured mitochondria release their contents into the cytoplasm, including cytochrome c. This step is closely related to apoptosis induced by the mitochondrial pathway [references to be included: .1-3 see below]. We have demonstrated that the absence of a prompt cardiac arrest results in mitochondrial swelling, which further leads to cell apoptosis. Thus, we concluded that mitochondria experience a different pathological process due to prompt cardiac arrest”.

1 - Alterations in mitochondrial structure and function are early events of dexamethasone-induced thymocyte apoptosis.

Petit PX, Lecoeur H, Zorn E, Dauguet C, Mignotte B, Gougeon ML. J Cell Biol. 1995 Jul;130(1):157-67. doi: 10.1083/jcb.130.1.157.

2 - Reduction in mitochondrial potential constitutes an early irreversible step of programmed lymphocyte death in vivo.

Zamzami N, Marchetti P, Castedo M, Zanin C, Vayssière JL, Petit PX, Kroemer G. J Exp Med. 1995 May 1;181(5):1661-72. doi: 10.1084/jem.181.5.1661.

3 - Disruption of the outer mitochondrial membrane as a result of large amplitude swelling: the impact of irreversible permeability transition.

Petit PX, Goubern M, Diolez P, Susin SA, Zamzami N, Kroemer G. FEBS Lett. 1998 Apr 10;426(1):111-6. doi: 10.1016/s0014-5793(98)00318-4

Remark 2

Figure 1:  the a part could closer to the b and c part of the figure (the blank could be reduced). This will represent again of space, the figure becoming more compact.

Remark 3

Figure 3, same remark (need small compaction)

Remark 4

Indeed, all figure needed small compaction and blank spaces deletion

Reviewer 2 Report

Wu et al, compared mice with complete and prompt cardiac arrest with mice without using cardioplegic solution on graft protection and functional improvement. I believe cardioplegic solution is routinely used in heart transplantation in clinic. Thus, the concept is not new. Although authors found that activated mitophagy may be involved, the originality of this paper is low. 

Round 2

Reviewer 2 Report

As  I mentioned in my previous report this method is routinely practiced in clinic, thus the novelty and translational potential is low. 
